# Nephroprotective Plants: A Review on the Use in Pre-Renal and Post-Renal Diseases

**DOI:** 10.3390/plants11060818

**Published:** 2022-03-18

**Authors:** Mario Adrián Tienda-Vázquez, Zoé P. Morreeuw, Juan Eduardo Sosa-Hernández, Anaberta Cardador-Martínez, Ernesto Sabath, Elda M. Melchor-Martínez, Hafiz M. N. Iqbal, Roberto Parra-Saldívar

**Affiliations:** 1Tecnologico de Monterrey, School of Engineering and Sciences, Monterrey 64849, Mexico; a01204468@itesm.mx (M.A.T.-V.); zpmorreeuw@tec.mx (Z.P.M.); eduardo.sosa@tec.mx (J.E.S.-H.); mcardador@tec.mx (A.C.-M.); 2Departamento de Nefrología, Hospital General de Querétaro, Queretaro 76175, Mexico; esabath@yahoo.com; 3Facultad de Ciencias Naturales, Universidad Autónoma de Querétaro, Juriquilla 76230, Mexico

**Keywords:** plants, pre-renal diseases, post-renal diseases, secondary nephroprotection

## Abstract

Kidney diseases are expected to become the fifth leading cause of death by 2040. Several physiological failures classified as pre-, intra-, and post-renal factors induce kidney damage. Diabetes, liver pathologies, rhabdomyolysis, and intestinal microbiota have been identified as pre-renal factors, and lithiasis or blood clots in the ureters, prostate cancer, urethral obstructions, prostate elongation, and urinary tract infections are post-renal factors. Additionally, the nephrotoxicity of drugs has been highlighted as a crucial factor inducing kidney injuries. Due to the adverse effects of drugs, it is necessary to point to other alternatives to complement the treatment of these diseases, such as nephroprotective agents. Plants are a wide source of nephroprotective substances and can have beneficial effects in different levels of the physiological pathways which lead to kidney damage. In traditional medicines, plants are used as antioxidants, anti-inflammatories, diuretics, and anticancer agents, among other benefits. However, the mechanism of action of some plants empirically used remains unknown and scientific data are required to support their nephroprotective effects. The present work reviewed the plants with a beneficial effect on kidney diseases. The classification of nephroprotective plants according to the clinical definition of pre-renal, intrinsic, and post-renal factors is proposed to orient their use as complementary treatments.

## 1. Introduction

The kidneys play an important role in human physiology, maintaining fluid homeostasis, regulating blood pressure, erythrocyte production and bone density, regulating hormonal balance, and filtering and removing nitrogenous and other waste products [1,2]. Chronic kidney disease (CKD) is characterized by a progressive loss of functions while acute kidney injury (AKI) is an abrupt reduction in kidney function. Both CKD and AKI have increased worldwide and are considered one of the leading public health problems [2,3]. A projection of health concerns by 2040 ranked CKD as the fifth leading cause of death worldwide [3]. In addition, kidney diseases have been recognized as risk factors for severe forms of COVID-19 [4]. The increase of their prevalence is associated with the increase of diabetes Mellitus (DM) and hypertension as the main reported causes of kidney dysfunctions, although various factors can trigger this physiopathology. Factors are classified, based on the pathway they led to kidney damage, as pre-renal, intrinsic, and post-renal factors (Figure 1).

According to clinical criteria, pre-renal diseases are related to a decrease in renal perfusion or alteration in the systemic circulation, which will first compromise the glomerular filtration rate (GFR) and secondly lead to more severe alterations in the kidney structure. These dysfunctions are reflected in clinical analyses by changes in biomarker levels; for example, an increase in serum creatinine; and fluctuations in urine flow [5,6]. Several diseases have been identified as pre-renal factors such as bleeding, trauma, shock, hypertension, cirrhosis, diabetes, systemic infections, hypotension, autoimmune diseases, rhabdomyolysis, disorders of the gut microbiota, liver damage, and intravascular volume depletion [7]. The factors that directly cause kidney damage are intrinsic, whether heavy metals, trauma, Wegener’s granulomatosis, proteinuria, congenital abnormalities, drug toxicity, renal atheroembolic disease, arthralgias, lupus erythematosus, or kidney cancer. Histologically, the main diagnoses are ischemic acute tubular necrosis, nephrotoxic acute tubular necrosis, and glomerulonephritis [8]. Any other health disorders which can indirectly induce kidney failures that occur after the physiological action of the kidney are post-renal diseases. Among these post-renal factors, mainly related to urinary flow disorders, are the ureter and urethra obstruction due to blood clots, lithiasis, or tumor growth, which can lead to increased pressure inside tubules, and as a consequence, the GFR is compromised, and a urinary tract infection (UTI) [9] is caused.

Kidney diseases can be addressed at several levels according to the physiological pathway of the original cause. Each pre-renal and post-renal disease currently has pharmacological treatments; however, most of the drugs used cause adverse effects and sometimes lead to intrinsic kidney damage. Among them are non-steroidal anti-inflammatory drugs (NSAID), proton pump inhibitors, antibiotics, and chemotherapy [10,11,12,13]. Nephrotoxicity of drugs administrated as treatments for pre-renal and post-renal diseases, as well as for other diseases, is now considered as a risk factor of acute and chronic kidney conditions. To avoid the adverse effects caused by medication, different alternatives have been sought to treat these pathologies.

Plants have been traditionally used as treatments for various diseases, among them several pathologies identified as pre-, intra-, and post-renal factors. The medicinal characteristics of plants have been attributed to their secondary metabolites, which can protect against pathogens or have important physiological benefits to prevent some diseases [14,15]. Plants provide a wide range of bioactive compounds which act as antioxidants, anti-inflammatory, diuretic, anticancer, and antimicrobial [16,17,18]. Further, nephroprotective agents from plants mitigate processes such as interstitial nephritis, altered intraglomerular hemodynamics, tubular necrosis, or glomerulonephritis [19]. Previous works have already reviewed the usages of plants and phytochemicals as nephroprotective agents providing an important understanding of how extracts or single compounds interfere with molecular pathways to mitigate kidney diseases [2,20]. However, they usually focused on intrinsic damage such as nephrotoxicity, omitting the pre-renal and post-renal factors, and to our knowledge, no classification of nephroprotective plants according to pre-, intra-, and post-renal diseases have been reported.

Furthermore, since it is complicated to replace the effect granted by drugs, nephroprotective plants have to be considered as a complement of treatments rather than substitutive treatments [21,22,23]. The beneficial complementarity of traditional and modern medicine for primary health care is promoted by the World Health Organization (WHO). The WHO reported that a large part of the world population is using plants products as their primary source of health care. While the lack of fundamental understanding about the mechanism of actions of medicinal plants has limited their uses worldwide, especially in countries where modern medicines dominate the health systems [24]. However, the usage of medicinal plants as a key component of complementary and alternative medicine has acquired renewed interest. As a success story, traditional Chinese medicine, based mainly on plants, is globally accepted and used at every level of the China health system; guidelines for recommendations and uses of plants have already been implemented by the government and academic institutions [25,26]. Similarly, ancestral medicines of India, such as Ayurveda, have proved their efficiency as treatments of several diseases and play an important role in the economy of the Indian health sector [27,28]. On the other hand, nearly 25% of modern medicines are derived from natural products, many of which were first used in traditional medicines. Thus, plants and products from traditional medicines are a resource for primary health care, but also for innovation and discovery [26,29,30].

In this context, plants used as nephroprotective agents for pre-renal, intrinsic and post-renal disease were reviewed. Scientific information on the effect of the nephroprotective plants on the pathophysiologies that lead to kidney injuries was described to support their uses as a complementary pharmacological treatment.

## 2. A Combined Therapeutic Approach

Medicinal plants are used for wellbeing in every ancient civilization, and some of the traditional uses of plants are still employed, such as in traditional Chinese medicine and Ayurveda. Traditional usage of plants as treatments is based on knowledge shared over generations. [26,31,32,33,34,35,36]. In contrast, occidental medicine, in some cases, is based on a lifetime intake of drugs; therefore, patients become dependent on drugs just to survive, especially older people. Despite their therapeutic effects, drugs also have a wide range of adverse effects such as nephrotoxicity [37]. In addition, drugs can even generate abstinence syndrome, generating drug dependence in patients [38,39,40,41].

In the last 20 years, plants and their extracts have been used as healthy alternatives in the form of food supplements, herbal medicines, natural health products, phytomedicine, or dietary supplements [42]. It is now well-known that plants and their phytochemicals exert specific therapeutic effects for different health disorders, acting as antioxidants, antimicrobials, anti-inflammatories, diuretics, and anticancer agents, among other benefits. For renal pathophysiology, a plant-based diet has proven its benefits [43], in particular in a patient with mild proteinuria and diabetic nephropathy [44]. Likewise, herbal formulations help reduce the dialysis requirements by curing the causes and altering symptoms of renal failures and are also helpful in mitigating the side effects of dialysis [21]. Thus, the current approach of using plants as a complementary treatment provides a double benefit; they support the fight against the disease and at the same time act as nephroprotective agents.

The positive combined approach of occidental and traditional medicine raises new questions about plants and natural products for therapeutic uses. Indian and Chinese medicines are widely accepted due to their effectiveness, with fewer side effects reported in comparison to modern systems of medicine. Furthermore, plants in the management of diseases are presented as a cost-effective alternative [45,46]. However, there is still a lack of knowledge about how plants act as therapeutic agents. Further, effective doses for plant intakes as food or supplements as treatment are usually not established, and dose-related toxicity of plant chemicals is questioned [47,48,49].

## 3. Nephroprotective Plants

Now, thanks to advances in molecular techniques, scientific procedures, and protocols, the therapeutic effects can be demonstrated by in vitro or in vivo validated models and even clinical trials. These studies aimed to validate the benefits of plants and derivates, discarding plants that have a false or placebo effect. Perspectives are usually to recommend the use of plants as supplementary treatment or to develop naturally sourced drugs with lower toxicity than synthetic drugs.

To that end, key parameters of the studies are the origin of plants, the preparation of the plants as therapeutic products, the phytochemicals, the model of the study (clinical or in vivo), the model used to simulate the targeted disease, and the observed mechanism of action by the biomarkers analysis (Table 1). Previous reviews have already exhaustively listed the plants and their phytochemicals for health benefits. Hence, the present review focuses on the diversity of plants that have scientifically demonstrated their biological activities against renal associated disorders. The mechanisms of action of plant products are described below according to the physiological pathway they regulate. Plant extracts and compounds depicted in Table 1 are efficient against pre-renal or post-renal conditions and prevent intrinsic kidney damage. Plants that act first-line directly protecting the kidney were classified as intrinsic, and those that mitigate the disease or factor and at the same time protect the kidney from secondary damages were classified as pre-renal and post-renal. Based on clinical definitions, plants from these last categories could be called secondary nephroprotectors.

## 4. The Role of Plants in Renal Pathophysiology

### 4.1. Pre-Renal Factors

#### 4.1.1. Diabetes Mellitus

Diabetes mellitus (DM) is a disease characterized by hyperglycemia and insulin resistance. In type 1 diabetes, the pancreatic β-cells responsible for insulin production are compromised and cause an elevation in glucose level. Type 2 diabetes, is a low response of insulin secreted to the target tissues [72,73]. Over time, DM is a risk factor for CKD, and diabetic nephropathy is the most common cause of end-stage renal disease [45,79]. The pathophysiology induced by DM develops in multifactorial forms and can even trigger other renal risk factors.

Diabetes causes glomerular hyperfiltration and increases intraglomerular pressure due to the high amounts of glucose to filter. The sodium-glucose transport proteins (SLGT) and glucose transporter (GLUT) transporters in the proximal convoluted tubules have to overwork, leading to renal hyperperfusion. The tubules cannot handle the amount of glucose, and its excretion through the urine induces osmotic diuresis. Moreover, this can also cause microalbuminuria, macroalbuminuria, nephrotic syndrome, and chronic renal failure, gradually. Hyperglycemia also causes cellular dehydration by increasing the osmotic pressure of the extracellular fluid. Therefore, a hypovolemic state can be reached because of the loss of urine, due to a few reabsorptions, and intracellular dehydration. The kidney, through baroreceptors, detects low pressure and continuously activates the renin angiotensin aldosterone system (RAAS), leading to hypertension [80,81,82]. On the other hand, insulin resistance causes mitochondrial superoxide overproduction, activating protein kinase C (PKC) pathways via aldose sorbitol accumulation and the formation of advanced glycation end-products (AGEs). The reactive oxygen species (ROS) induces severe damage through lipid peroxidation, oxidizing low-density lipoprotein (LDL). ROS also activate inflammatory response activating the signaling pathway with the transcription factors Bcl2, NF-κB which promote the expression of inflammatory cytokines (IL-1β, IL-2, IL-6, IL-12 and IL-18, TNF-α and MCP1) and apoptosis cascade [45,80]. Finally, glomerular hypertrophy occurs with thickening of the basement membrane and mesangial expansion, which can lead to glomerulosclerosis, hemodynamic dysregulation, and tubulointerstitial fibrosis (Figure 2) [45,79].

Cocoa powder can help to counteract hyperglycemia by regulating glucose homeostasis and insulin resistance. The 10% cocoa diet for Zucker diabetic fatty rats restored the glucose transporters (SGLT-2 and GLUT-2) and prevented the inactivation of the glycogenesis regulating the GSK-3 (glycogen synthase kinase 3)/GS (glycogen synthase) pathway and phosphorylation (G-6-PASE: glucose 6 phosphatase). In addition, cocoa reverses the decrease of the phosphorylated levels of tyrosine-phosphorylated insulin [58]. Aqueous extract of *Coffea arabica* pulp, rich in polyphenols, efficiently prevents hyperglycemia, insulin resistance, and lipid metabolism disorders. The coffee pulp extract raised levels of the liver antioxidant enzymes catalase (CAT) and copper-zinc superoxide dismutase (Cu-Zn SOD), blocked the stress-sensitive signaling pathway by reducing the expression levels of p-PKCα/PKCα, and improved cationic transport. Such effects were observed with the coffee extract administrated as supplements at 1000 mg/kg body weight (BW) in a model of type 2 diabetes in rats, induced by a high-fat diet, and the effect was compared with metformin as antidiabetic treatment (30 mg/kg BW), and a coffee pulp extract/metformin combined treatment (1000/30 mg/kg BW) [59]. Likewise, in Wistar rats with renal damages caused by diabetes and the nephrotoxic drug streptozotocin, the use of leaf aqueous extract of *Anchomanes difformis* reverses tissue damage of mesangial cells, glomerular hypertrophy, and membrane damage. At the molecular level, the extract reduced the serum concentration of urea, reduced oxidative stress by increasing the levels of CAT and SOD, and had anti-inflammatory effects by reducing the expression of NF-κB and Bcl2, thus, decreasing the IL-10, IL-18 and TNFα, IL-18 levels. In this work, no significant difference was found between the two tested concentrations: 200 and 400 mg/kg BW, and the effect was not specifically associated with phytochemicals [45]. In contrast, similar antioxidant effects of *Hibiscus sabdariffa* infusion in the metabolic syndrome rat model have been attributed to the high content of antioxidant phytochemicals such as polyphenols, anthocyanins, flavonoids and phenolic acids. In addition to enhanced antioxidant physiological pathways, phenolic compounds participate in neutralizing ROS by donating a hydrogen atom or an electron. The beneficial effect on body weight, lipid metabolism, insulin homeostasis, and renal function was obtained with a treatment of 2% of *H. sabdariffa* infusion in drinking water [61]. Additionally, isolated flavonoids from *Eysenhardtia polystachya* have been tested for their antidiabetic and nephroprotective effects in diabetic mice with renal damages induced by streptozotocin. The results revealed that 20 mg/kg BW of the purified extract significantly reduced oxidative damages in both the kidneys and the liver, and such effect has been related to the high antioxidant capacity of the characterized phytochemicals [60]. In addition to polyphenols, antioxidative effects, and mitigation of serum lipid abnormalities observed in diabetic Wistar rats treated with *Agave lechuguilla* by-product extracts were attributed to the saponins (triterpenoid glycosides), and the effective concentration was established at 300 mg/kg BW [83].

The use of plants as a supplementary treatment for diabetes and renal protection is supported by the reviewed studies. A common factor of the plants to counteract the effects of diabetes-related pathologies is their antioxidant and anti-inflammatory properties. Such bioactivities have been mostly attributed to phenolic compounds. However, others phytochemicals can also be responsible for the observed effects and further targeted studies are required to elucidate their therapeutic potential. On the other hand, dose responses studies need to be carried out to establish the posology.

#### 4.1.2. Hypertension

Hypertension is a disease characterized by permanent or continuous high pressure; according to international guidelines on hypertension, systolic blood pressure values must be higher than 140 mmHg and diastolic blood pressure higher than 90 mmHg. However, pharmacological treatments depend on the patient’s age and comorbidities [84,85,86,87]. The kidney has self-regulating mechanisms to dampen fluctuations in systemic blood pressure to avoid an increase in intraglomerular pressure. However, at constant high blood pressure, these kidney mechanisms fail because the incoming vessels become weaker, stiffer, or thinner, and a phenomenon called myogenic reflex occurs. During the first compensatory response, the depolarization of the membrane, by increasing the intracellular calcium flux through the L-type channels, causes a change in caliber (contraction) in the afferent arteriole. When this first mechanism fails, there is an increase in intraglomerular filtration and pressure, also associated with an increase in the sodium chloride load. To compensate, another autoregulatory mechanism called tubuloglomerular feedback switches on, which is detected in the distal tubule by the macula densa cells. The failure in compensatory self-regulatory mechanisms ended in glomerulonephritis, the development of glomerulosclerosis, and a rupture of the fenestra, leading to the filtration of larger molecules such as proteins, which leads to proteinuria. As proteins accumulate in the tubules, they activate profibrotic, proinflammatory, and cytotoxic pathways, causing tubulointerstitial damage and kidney scarring (Figure 3) [88,89,90].

In this context, tincture obtained from the herbaceous plant *Cichorium intybus* showed promising cardioprotective and nephroprotective effects in the rat isoprenaline-induced myocardial ischemia model. Therapeutic effects were evidenced by the increased antioxidant and anti-inflammatory mechanisms activity and the decreased creatinine kinase myocardial band (CK-MB), aspartate aminotransferase (AST), and malondialdehyde (MDA) levels. Such effects have been related to the high antioxidant capacity determined in vitro and attributed to the polyphenolic acids and flavonoids quantified in the extracts. The higher nephroprotective effect was obtained with 100 mg/mL in drinking water compared to lower concentrations, highlighting that higher concentrations may have adverse oxidant and inflammatory effects [62]. Similarly, the anti-hypertensive effect of polyphenol-rich *H. sabdariffa* infusion has been demonstrated in humans with uncontrolled hypertension. Proof of effect has been obtained with doses ranging from 10,000 to 20,000 mg/d, however, the authors hypothesized that a lower dose could be sufficient, from 2500 to 5000 mg/d, and could help to minimize the probability of gastric acidity that has been observed as a side effect in very few cases [46]. The mechanism of blood pressure regulation has been further described in spontaneously hypertensive rats and Wistar-Kyoto rats supplemented with 360 mg/kg BW of *Gardenia jasminoides* ethanolic extracts or 25 and 50 mg/kg BW of the purified active compounds geniposide (terpene iridoid glycoside). This work showed that geniposide decreased the left ventricular end-diastolic diameter (LVEDD) and left ventricular end-systolic diameter (LVESD), and improved the systolic function by increasing the left ventricle ejection fraction (LVEF) and left ventricular fraction shortening (LVFS). At a molecular level, the analysis of myocardial injury biomarkers revealed that geniposide treatment enhanced cardiac function by activating the energy metabolic pathway (AMPK/SirT1/FOXO1), and decreased apoptosis rate by regulating the p38/Bcl2/BAX pathway [63]. The effects of *G. jasminoides* have been only partially explained by the geniposide; thus, other bioactive compounds must also have cardioprotective benefits and have to be characterized. From another perspective, the synergic effect of phytochemicals on plants extracts could potentialize the therapeutical effect.

The good results obtained, particularly in the human trial, encourage further studies on the anti-hypertensive properties of plants to determine effective doses, characterize the bioactive phytochemicals, evaluate the potential side effects, and related their uses with nephroprotective effects.

#### 4.1.3. Hepatic Injury

The liver is the main organ in charge of metabolizing xenobiotics, and metabolic processes such as hydroxylation, conjugation, acylation, reduction, oxidation, sulfonation, and glucuronidations [50]. The main causes of liver damage are high doses of non-steroidal anti-inflammatory drugs, alcohol consumption, leptospirosis, infections, acetaminophen poisoning, antibiotics, and viral hemorrhagic fever [91,92].

Liver disease has 4 different stages; stages 1 and 2 belong to the compensatory phase characterized by being asymptomatic, whereas stages 3 and 4 belong to the decompensatory phase, characterized by ascites, variceal bleeding, hepatic encephalopathy, ending in the last stage in sepsis and kidney failure. Cirrhosis is representative of the last stage of chronic liver disease, commonly called cirrhosis, characterized by regenerative nodules, progressive fibrosis, and chronic inflammatory response, leading to hypertension. Multiple mechanisms that cause acute kidney damage, secondarily to cirrhosis, have been observed. An alteration in the liver circulatory system as a consequence of chronic inflammation and hypertension results in an excessive release of vasodilators such as carbon monoxide (CO) and nitric oxide (NO). The foregoing decreases vascular resistance causing heart failure, which is initially compensated by an increase in heart rate. As the disease progresses, GFR decreases, leading to the activation of endogenous vasoconstrictor systems such as the sympathetic nervous system (SNS), endothelin (ET), arginine vasopressin (AT), RAAS, thromboxane A2 and leukotrienes, which in turn causes edema and ascites (Figure 4). Secondary to the above, there is an increase in intestinal permeability, causing bacterial translocation, systemic inflammation, and oxidative stress [91,93].

The model of liver damage induced by carbon tetrachloride (CCl_4_) is characterized by lipid peroxidation and subsequent MDA. In addition, the CCl_3_O_2_• radical interacts with the polyunsaturated fatty acids of the endoplasmic reticulum which is reflected in high concentrations of bilirubin, serum glutamic-pyruvic transaminase (SGPT), serum glutamic oxaloacetate transaminase (SGOT), and alkaline phosphatase (ALP). Over time, such metabolic alteration generates necrosis of liver tissue. In the CCl_4_-induced Wistar rat model, methanol extract of the medicinal plant *Tinospora crispa* significantly moderated the elevation of all the biomarkers and improved antioxidant response increasing the levels of SOD. The maximum hepatoprotective activity was observed at 400 mg/kg BW, which was the highest concentration tested in the trial. Among the phytochemicals characterized in the extract, the in silico prediction of activity spectra for substances suggested that the diterpenoid tinocrisposide has the highest hepatoprotective potential. However, the computer-aided pharmacodynamic analysis revealed that this compound is not a suitable drug candidate, in fact, only the flavonoid genkwanin appeared as a safe hepatoprotective natural product drug. This study highlighted the paradox between therapeutic property and potential toxicity of the phytochemicals [50]. In the same model, nonpolar extracts of the flowers *Cirsium vulgare* and *Cirsium ehrenbergii* showed comparable hepatoprotective effects with a dose-dependent response evidence between 250 and 500 mg/kg BW. The main molecule found in the extracts is lupeol acetate (triterpenoid), thus they assumed it is the protective agent. The study suggested that lupeol acetate has antioxidant properties avoiding damage caused by oxidative stress; it inhibits pro-inflammatory enzymes and prevents glycogen depletion [51]. Although such therapeutic properties have to be confirmed by evaluating isolated lupeol acetate effect in vivo or predicting its activity through in silico PASS and pharmacodynamic analysis. The aqueous-ethanolic extract of the edible halophyte plant *Suaeda vermiculata,* with high content of antioxidant flavonoids, proved to capture free radicals generated by CCl_4,_ this extract achieved a decrease in AST and ALT, in male Sprague Dawley rats. In addition to the hepatoprotective, nephroprotective, and cardioprotective effects demonstrated at 250 mg/kg BW, the safety of the extract was confirmed at 5 g/kg BW, and IC_50_ was established at 56.19 and 78.40 µg/mL by in vitro assay. Furthermore, the IC_50_ of the cytotoxic drug doxorubicin was 2.77-fold reversed when co-administered with the *S. vermiculata* extracts, this effect was defined as synergetic [52]. Likewise, curcumin powder (polyphenol compound obtained from the rhizome of the *Curcuma longa*) protected the kidney from damage caused by doxorubicin in Wistar rats at 200 mg/kg BW. The biomarkers analysis revealed that curcumin increases antioxidant enzyme activity (GPx (glutathione peroxidase), CAT, and SOD), prevents lipid peroxidation, reduces inflammation by modulating cytokine levels (TNF-α, NF-κB, IL-1β, iNOS, and COX-2), and mitigates toxicity by limiting apoptosis activation (caspase 3) [65].

Another type of hepatotoxicity is acetaminophen-induced toxicity (APAP) which is due to the production of the reactive metabolite *N*-acetyl-p-benzoquinone imine (NAPQI) through sulphation and glucuronidation metabolic pathways. At high concentrations, liver GSH is overwhelmed and NADQI, which is not scavenged reacts with mitochondrial proteins of the hepatocytes. Mitochondrial damage increases oxidative stress and subsequently leads to hepatocyte necrosis [92]. Moreover, nitric oxide (NO) levels rise in the proximal convoluted tubules of the kidney, the glomerulus, and distal convoluted tubules; this vasodilator alters the kidney circulatory system. In a Swiss albino mice study with acetaminophen-induced damage, the co-administration of *Descurainia sophia* seed extract at 300 mg/kg was significantly protected from nephrotoxicity. Proximal convoluted tubule structure was preserved, inflammation, swelling, and necrosis were reduced, and lower levels of uric acid, creatinine, and blood urea nitrogen (BUN) were observed [53].

In another toxicity model, thioacetamide (TAA) recreate acute liver injury and cirrhosis similarly as with CCL_4_ and APAPA. When it is metabolized, the highly reactive thioacetamide-S-dioxide is produced with subsequent elevation of oxidative stress and inflammatory cytokines levels [94]. In male Sprague-Dawley rats thioacetamide-induced hepatotoxicity and nephrotoxicity were successfully mitigated by an herbal extract. The high content of antioxidant polyphenols in *Euphorbia paralias* extracts improved the redox status of the kidney tissue, reducing the serum creatinine, serum urea and increasing CAT and SOD levels. The histological analysis revealed that the extract, administrated at 200 mg/kg body weight, efficiently prevented damage to the nephron of blood vessel congestion and glomerular damage [67].

Hepatotoxicity and subsequent nephrotoxicity can also be induced by non-steroidal anti-inflammatory (NSAID) drugs such as paracetamol, one of the most consumed analgesics worldwide. In a study of the paracetamol-induced toxicity model in Wistar rats treated with *Eurycoma longifolia*, protection was observed by decreasing levels of creatinine, urea, albumin, and total serum protein. At a tissue level, it achieved the preservation of glomeruli, interstitium, and tubules in the kidneys. The medicinal herb extract presented an effect at 200 mg/kg BW, with a dose-dependent effect observed when increased at 400 mg/kg BW [54]. Similarly, passion fruit (*Passiflora* spp.) peel extract maintained the biomarkers serum renal function such as urea and creatinine at normal levels when co-administrated with paracetamol in albino rats. The authors also highlighted a dose-dependent nephroprotective activity from 150 to 500 mg/kg BW. This effect has been associated with the antioxidant potential of flavonoids and tannins as the main phytochemicals found in the extracts [66].

When pre-renal diseases reach their final stages, damages are irreversible, and organ transplants are required. Cyclosporine-A is used as an immunosuppressive drug to enhance transplantation success, although overdoses of this molecule induce organ damage. In albino male Wistar rats with Cyclosporine-A-induced nephrotoxicity, the leaf extract of medicinal plant *Costus afer*, at 200 mg/kg BW, decreased serum potassium and blood urea nitrogen levels, inhibited the elevation of MDA, increased antioxidant defense, and prevented any structural change (glomerular and tubular histology) [69].

Hepatic injuries are mainly due to toxic substance ingestion, and hepatotoxicity and nephrotoxicity are closely related. The protective effect of the plant products is reported from 200 mg/kg BW with a dose-dependent response. Such therapeutical potential of plants, in several toxicity models and combined with widely used drugs, supports their use to treat pre-renal disease as well as to prevent secondary renal damages. The mentioned work underlined the action mechanisms of plants products, although, further analysis is necessary to accurately relate the observed effect with phytochemicals, which could be through targeted in vivo analysis or new in silico approaches.

#### 4.1.4. Antibiotic Drugs Damage

Additional risk factors are infections that also can be generated by multi-resistant bacteria; in these cases, the use of certain antibiotics such as polymyxins and colistin should be considered as a last resort. Plants are widely recognized as a source of antibacterial agents and are of interest for being active against antibiotic-resistant strains. For instance, crude extracts and solvent fractions of leaves and stems of the medicinal plant *Arbutus pavarii* have been evaluated on methicillin-resistant *Staphylococcus aureus* (MRSA) strains. The in vitro assays revealed that all the extracts and fractions exerted bacteriostatic and bactericidal effects. The metabolite profiling suggested that phenolic acids and flavonoids, as the main phytochemicals in the extracts and fractions, are responsible for the antibacterial activity [55]. Antimicrobial properties of plants have been widely reported, although the majority of the study is conducted in vitro, and results have to be confirmed through in vivo assays to further recommend their uses as treatment of infectious diseases. That could explain why antibiotics are still largely used.

Antibiotics can lead to hepatotoxicity and nephrotoxicity. The toxic effect is generated by anchoring in the membrane of the proximal tubule; on the edge of the brush, there are negative charges and these antibiotics have a polycationic ring in their structure, for subsequent internalization and cellular damage. Additionally, it upregulates cholesterol biosynthesis and increases urinary cholesterol levels. The nephrotoxicity of gentamicin follows an anchorage in the same area of the proximal tubule, subsequent internalization by endocytosis, rupture within the cell, the release of proteases, damage to the organelles, generation of ROS, ending in necrosis [56,95]. Some antibiotics could cause AKI through mitochondrial injury with subsequent ROS production and change in metabolic energy consumption pathway, alterations in renal circulation at micro and macro levels, and tissue damage [13]. The use of antibiotics is essential in daily life to fight infections, which, if not treated, could cause sepsis shock; here it can be observed that fighting these diseases with antibiotic drugs also causes hepatotoxicity and nephrotoxicity. In vivo, plants have proved their potential in reverting antibiotic-induced damages in liver and kidney tissues. In Wistar rats treated with gentamicin and Atlas mastic tree (*Pistacia atlantica*), leaf extracts simultaneously showed lower antibiotic-induced nephropathy with a dose-dependent response evidenced between 200 and 800 mg/kg body weight. The protective effect was attributed to the antioxidant and anti-inflammatory effect of the phenolic acids and flavonoids. The reduction of inflammation was evidenced by a decrease of the serum lipid profile level and an increase of high-density lipoproteins level (HDL). Protective effects against oxidative damage were reflected in the reduction of MDA prevalence by increasing the plasma antioxidant capacity with higher activity of CAT and SOD and higher vitamin C level [68]. Similarly, in the same gentamicin-induced nephropathy model, the leaf extract of *Punica granatum* (pomegranate) decreased serum creatinine, urea, and albumin levels as well as urine albumin. In addition, this extract eliminated hydroxyl radicals and singlet oxygen, increased the number of antioxidant enzymes such as CAT, SOD, and GSH, decreased MDA and expression of TNF-α, and lastly, in the tissue it improved morphological alterations such as tubular atrophy, necrosis, vascularization and congestion of peritubular blood vessels. Such effect was demonstrated at 200 and 400 mg/kg BW, whereas at 100 mg/kg BW, incomplete nephroprotection was obtained [56]. In contrast, 100 mg/kg BW of pomegranate fruit peel extract showed effective hepatoprotective and nephroprotective properties in co-treatment with high doses of the antibiotic vancomycin, and better protection was highlighted when administrated prior to vancomycin treatment [57]. This result suggests an antagonist effect of the antibiotic and the plant extract.

Hence, plants and their different parts can be used to reverse antibiotic toxicity not only in the kidney but also in the liver and gut, particularly acting as antioxidants and anti-inflammatory. The strategy for complementary treatment such as the mode of administration and doses must be further studied to guarantee the therapeutic effect of both plant products and antibiotic drugs.

#### 4.1.5. The Gut Microbiota

The intestinal microbiota studies have gained importance since it has been proven that its alteration leads to the production of uremic retention solutes (URS) and is directly related to a deterioration in kidney function. One of these toxic metabolites is trimethylamine *N*-Oxide (TMAO). The TMA molecule is produced by the microbiota from its dietary precursors such as carnitine, choline, and betaine obtained mainly from animal protein intakes. Later, in the liver, it is oxidized thanks to monooxygenase, released into circulation, and reaches the kidneys, in this part, the kidneys have to work to excrete the metabolite. TMAO increases endogenous inflammation, promotes atherogenesis, and modulates lipid metabolism. It has been shown in vivo studies and clinical trials that the intake of vegetable protein decreases the TMAO levels [96,97], supporting the benefice of a plant-based diet and the use of plant supplements to treat pre-renal diseases.

Antidiabetics, antibiotics, analgesics and antipyretics, and other drugs, in addition to causing liver and kidney damage, are also responsible for alterations in the intestinal microbiota, causing diarrhea, among other physiological disorders. In diarrhea, the decrease in probiotics is affected and there is an overgrowth of opportunistic pathogens. One of the most common usages of plants as complementary pharmaceutical treatment is as prebiotics. Several phytochemicals have already proved to positively modulate gut microbiota, enhancing probiotics growth and limiting pathogens developments. Among them, polyphenol resveratrol is a compound synthesized by a high diversity of plants. Due to its low bioavailability, it is not early metabolized thus reaches the colon and interacts with the gut microbiota, changing the composition of the microbial community. By changing the microbiota, tight junctions can be increased to form a barrier that prevents harmful metabolic waste from crossing and arriving at the liver; this interaction is called the gut-liver axis. Resveratrol (50 mg/kg BW) repaired the tight junction in non-alcoholic fatty liver disease induced by the high-fat diet in C57BL/6J mice model. It also increased *Olsenella* and *Allocaculu* genus, which exhibit a beneficial change for the disease [98]. In a C57BL/6 mouse model with lincomycin hydrochloride-induced diarrhea, several medicinal herb residues (*Dioscorea opposita* rhizome, *Pseudostellaria heterophylla* root tuber, *Crataegus pinnatifida* fruit, *Citrus reticulata* pericarp and *Hordeum vulgare* fruit) fermented with probiotics (*Bacillus subtilis*, *Aspergillus oryzae* and *Lactobacillus plantarum* M3) were tested for their beneficial effect on gut microbiota. The fermentation supernatant significantly inhibited diarrhea caused by the antibiotics, and enhanced bacterial diversity and restored *Lactobacillus johnsonii* dominance in the gut microbial community. Furthermore, antioxidant and antibacterial properties were demonstrated in vitro [99]. In this last referenced work, the authors encourage the use of medicinal herb residues previously processed by pharmacological firms to obtain new therapeutical products. This highlights that the potential of plants in pharmacology is far from being fully exploited.

#### 4.1.6. Rhabdomyolysis

Rhabdomyolysis is a syndrome characterized by muscular sarcolemma injuries. Two pathways have been identified, failure in energy production by sodium-potassium ATPase and calcium ATPase pumps, and the activation of calcium-dependent phospholipases and proteases by an increase in intracellular calcium. These enzymes destroy the membrane and cytoskeleton proteins causing necrosis. Because of necrosis, electrolytes and intracellular proteins such as myoglobin, creatine kinase, lactate dehydrogenase, aspartate transaminase and aldose are released into the systemic circulation. Rhabdomyolysis syndrome is mainly caused by metabolic, genetic, structural, inflammatory and/or traumatic causes such as crush syndrome, muscular hypoxia, intense exercise, genetic defects, drug and/or medication abuse [100,101]. In addition to these factors, there is an association with antibiotics such as cefditoren, daptomycin, cefaclor, norfloxacin, erythromycin, clarithromycin, azithromycin, meropenem, cefdinir, trimethoprim-sulfamethoxazole, piperacillin-tazobactam, linezolid and ciprofloxacin [102].

The rhabdomyolysis subsequently causes kidney damage through the activation of platelets and the heme group (product of muscle necrosis); this group interacts with macrophage antigen 1 (Mac-1) and promotes citrullination of histones, ROS production, and subsequent macrophage extracellular trap (MET) formation. Kidney damage occurs through damage to cells of the proximal convoluted tubule due to the accumulation of ROS, lipid peroxidation, and precipitation of myoglobin with uromodulin (Figure 5) [100,103].

Because this disease must be treated carefully, it is important to take extra care of the kidney, avoiding any adverse effects from medications. For example, some medicinal herbs (*Pteridium* sp.) have been investigated for being responsible for rhabdomyolysis and multiple organ dysfunction in patients with no particular medical history and one with hypertension. This plant contains flavonoids, cardiac glycosides, saponins and phenols; however, the toxicity could not be attributed to one phytochemicals in particular [104]. In contrast, the effects exerted by curcumin have been presented as a promising option for the management of rhabdomyolysis. In a glycerol-induced rhabdomyolysis model of C57BL/6J mice, curcumin reduced ROS production by the activation of the Nrf2/HO-1 axis, reversed the decrease of renal GSH levels, and reduced the activation of NF-κB and ERK pro-inflammatory pathways. Moreover, histopathology showed that curcumin improved tubular cell death and lumen dilatation, interstitial edema, and loss of brush border. Such effects were obtained using 1000 mg/kg BW of curcumin as preventive treatment and after rhabdomyolysis induction. Furthermore, HO-1 was identified as a key pathway involved in the nephroprotective effect of curcumin [64]. The use of plants to prevent renal injuries requires specific attention when the pre-renal factor is rhabdomyolysis due to the potential adverse effect of some plants’ phytochemicals. In this context, purified extracts and compounds must be preferred rather than complex extracts to avoid negative effects and provide a therapeutic alternative.

### 4.2. Post-Renal

#### 4.2.1. Urinary Tract Infections

Urinary tract infections (UTI) are common ailments, mainly caused by bacteria such as Escherichia coli, Klebsiella pneumoniae, Proteus mirabilis, Enterococcus faecalis, Staphylococcus saprophyticus, Proteus spp., Streptococcus agalactiae, and Pseudomonas aeruginosa. When the pathogens are positioned in the bladder and prostate, they cause cystitis and prostatitis respectively, being classified as lower UTIs. When the uropathogens reach the kidney, they are classified as upper UTIs. Renal damage caused by UTI is characterized by severe inflammation in the interstitial tubule and subsequent fibrosis, which is called pyelonephritis (Figure 6). During fibrosis, kidney scars are created; due to this non-functional tissue, they cause a decrease in kidney activity, which eventually leads to the development of chronic kidney disease [105,106,107].

Different herbal strategies have been used to avoid the use of antibiotics and also combat this type of infection. Diuretic plants are widely recommended in case of UTI to help in eliminating uropathogens from the organism, thus, preventing secondary renal infection. The diuretic effect of *Equisetum arvense* (Field Horsetail) claimed in the traditional use of this herbaceous plant was confirmed through a randomized, double-blind clinical trial, with 900 mg/d of a dry extract containing 0.026% of flavonoids was administrated to healthy male volunteers [73]. At the same dose, the dry cranberry extract was administrated to male and female volunteers and resulted in significant inhibition of *E. coli* adhesion in male urine analyzed ex vivo while no significant effect was found with female urine. The nephroprotective effect was demonstrated in vitro with a decrease of bacterial adhesion in human A498 kidney cells. In addition to the antimicrobial properties of the phenol identified in cranberry hypothesized that the antiadhesive effect is due to endogenous compounds produced by the human organisms. They observed an increase in THP level in urine, this glycoprotein is a strong inhibitor of type 1 fimbriae adhesion, although such response was only found in male urine [71]. In another work, the use of propolis potentiates the effect of cranberry extract, reducing bacterial motility and biofilm formation in vitro regardless of the resistance of the tested strains to antibiotics [70].

The two clinical trials demonstrated the potential of diuretic plants as a preventive treatment for UTI and subsequent renal infections. However, the gender-specific physiological response highlighted provides new insight into the mechanism of action of plants as therapeutic agents and questions the use of male models as default in the in vivo assays to underlying physiological processes and develop new pharmacological treatments.

#### 4.2.2. Urinary Tract Obstructions

Lithiasis is the term used to refer to kidney stones; this could be triggered by low activity, diet, or genetics. The supersaturation of crystals such as calcium oxalate, magnesium phosphate, ammonium, uric acid, and cystine lead to stones formation. In some cases, stones could cause infections, pain, obstruct the flow, and hemorrhage (Figure 7) [108]. Stone prevalence is increased in case of hypercalciuria, hyperoxaluria, hypocitraturia, UTI, and low urinary volume. The probability of flow obstruction increases with the size of the stones [109]. Obstruction drastically reduces renal function, causing a decrease in glomerular filtration rate and contraction of afferent arterioles. In addition, oxalate delivers free radicals that can lead to lipid peroxidation and subsequent tissular changes. Lithiasis induced by ammonium chloride with ethylene glycol in a male Wistar rat model was successfully treated with an aqueous extract of *Descurainia sophia* seeds, a plant used for its diuretic property. The extract prevented tissue damage by decreasing calcium oxalate deposits in collecting ducts and urinary space. The dose-dependent response between 200 and 400 mg/kg BW was observed for some of the anatomic parameters, and lower calcium oxalate deposition was found for 400 mg/kg BW. The mechanism of action of the plant as an anti-lithiasis and nephroprotective agent is not further described, however, the effect was associated with the high content of flavonoids, fatty acids, and mucilages [72].

#### 4.2.3. Prostate Enlargement

Increased prostate volume can lead to urinary flow disorder and urethral obstruction therefore can cause subsequent renal damage (Figure 7). Prostate enlargement can be due to benign prostatic hyperplasia (BPH). Although the etiology is not yet fully defined, some theories agree on the conversion of testosterone to dihydrotestosterone (DHT) by the enzyme 5-α-reductase, to later promote the production of growth factors. DHT has a high affinity for androgen receptors, thus pharmacological treatments have been based on the use of 1-α-adrenoreceptor antagonists and 5-α-reductase inhibitors [77]. However, side effects have been reported such as erectile dysfunction, loss of libido, and dizziness, particularly for 5-α-reductase inhibitor drugs. As an alternative, 50 mg/kg of the *Cynanchum wilfordii* aqueous extract effectively avoided the negative effect of endogenous testosterone conversion in a rat model with BPH induced by testosterone. This beneficial effect was evidenced through the down-regulated expression of the gene encoding for the androgen receptor 5 α which consequently results in decreased testosterone and DHT levels. Two compounds, 4-hydroxyacetophenone and 2,4-hydroxyacetophenone, were identified as the chemical agents responsible for regulating gene expression [75]. In an open mono-central trial human trial, an oil-free pumpkin (*Cucurbita pepo)* seed hydroethanolic extract was used to treat BPH symptoms. The daily administration of a tablet formulated with 350 mg of crude extract induced a significant decrease in residual urine volume and nocturia after 12 weeks of treatment. The mitigation of BPH symptoms was attributed to the phytosterols and fatty acids as the active molecules from the pumpkin seeds [76,77].

On another hand, prostate cancer can generate similar disorders as BPH by increasing prostate volume due to tumor growth. Prostate cancer is one of the most prevalent cancers in men. Its conventional treatment is androgen deprivation therapy without taking chemotherapy into account [110]. This treatment based on the inhibition of androgens additionally causes kidney damage due to resulting hyperglycemia and dyslipidemia, which alters glomerular filtration [111]. Further, it can cause ischemic injury by attenuating vasodilation in the area of the renal vessels derived from testosterone and nitric oxide, and this type of injury cannot be prevented by estrogen, since it generates hypogonadism and estrogen levels decrease [112]. In addition, the growth of abnormalities in the epithelium of the prostate can obstruct urinary flow, leading to a decrease in GFR. In of 15% transgenic adenocarcinoma of the mouse prostate (TRAMP) model, 15% of broccoli sprout powder in the diet reduced the cancer incidence and progression. At a molecular level, broccoli influenced specific epigenetic mechanisms such as HDAC expression, the acetylation of histone (H3 lysine 18), the H3K9 trimethylation in the ventral prostate lobe, H3K18 acetylation levels, the mean area/foci for H3K9me3, the NAD(P)H quinone dehydrogenase 1 gene (NQO1) and p16 mRNA levels. The therapeutic effect was attributed to the sulforaphane found at 400 mg/kg in the formulated diet [78].

In the case of both BPH and prostate cancer, the mentioned studies promote the beneficial effect of plant-based diet and plants intakes as a supplement to prevent post-renal diseases and subsequent renal pathologies.

## 5. Conclusions and Future Perspectives

Kidney disease can be triggered by multiple factors or diseases, classified as pre-renal, intrinsic, and post-renal according to the physiological pathway via which they lead to kidney injury. The pharmacological treatments for the diseases raised in this review, in most cases, generate adverse effects and end in nephrotoxicity. In contrast, plants and their bioactive molecules offer nephroprotective effects, acting as antioxidants, anti-inflammatories, antibiotics, anti-cancer, and diuretics, among other effects. The reviewed works undoubtedly support the use of plants and their bioactive molecules to mitigate risks factors and drug-induced kidney injuries. Furthermore, these results provide a better understanding of how plants act as molecular modulators to alleviate pre-renal and post-renal disorders that indirectly or subsequently can lead to the development of intrinsic kidney disease. Indirect nephroprotective effects have been demonstrated from a wide diversity of plants, including all parts (roots, herbs, leaves, flowers, fruits, and seeds), as well as their by-products such as medicinal plant residue, fruits peels, and pulps. Likewise, therapeutic effects were reported from raw material, crude extracts, and purified compounds. Tested plant products provided nephroprotection at a concentration up to 200 mg/kg body weight in male rat models and 900 mg/d in human experiments. In general, no toxicity was evidenced at the highest tested concentration. The observed effects were mostly attributed to phenolic compounds, although few of the studies quantified and characterized the other phytochemicals.

The main perspective that can be highlighted from this review is that it broadens the panel of nephroprotective plants by considering not only plants with intrinsic effects or that can reverse nephrotoxicity, but also plants with beneficial effects on pre- and post-renal diseases. Therefore, a new classification of secondary nephroprotective plants is proposed.

Furthermore, to implement the use of plants and products as complementary treatments rather than substitutive treatment, key parameters must be further studied such as the ratio for co-administration of nephroprotective plant agents and nephrotoxic drugs, the dose-dependent toxicity, and qualitative and quantitative characterization of phytochemicals in the plant products. To achieve this, novel tools such as in silico pharmacodynamics can be particularly useful for identifying new nephroprotective agents and specifically designing in vivo and clinicals assays. In addition, both in vivo and clinical trials should be improved by considering female models.

## Figures and Tables

**Figure 1 plants-11-00818-f001:**
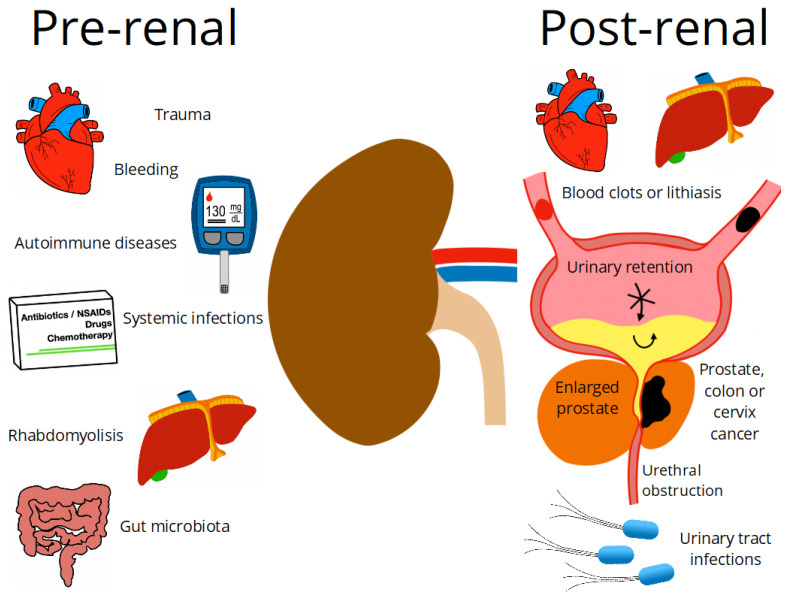
Pre-renal and post-renal diseases.

**Figure 2 plants-11-00818-f002:**
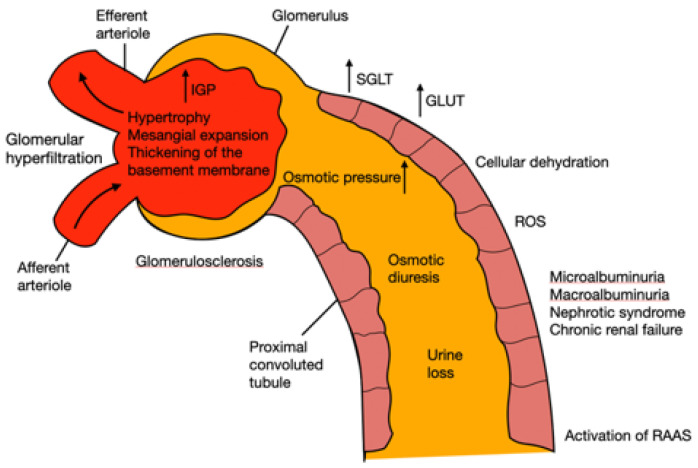
Pathophysiology of kidney damage induced by diabetes. IGP: intraglomerular pressure; SGLT: sodium-glucose transporter; GLUT: glucose transporter; RAAS: renin angiotensin aldosterone system.

**Figure 3 plants-11-00818-f003:**
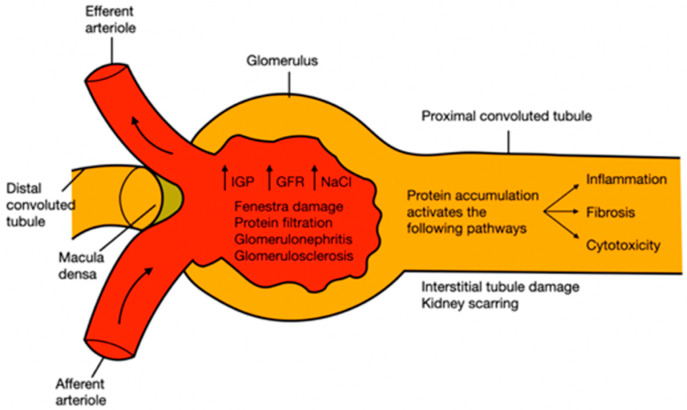
Pathophysiology of kidney damage induced by hypertension. IGP: intraglomerular pressure; GFR: glomerular filtration rate; NaCl: sodium chloride.

**Figure 4 plants-11-00818-f004:**
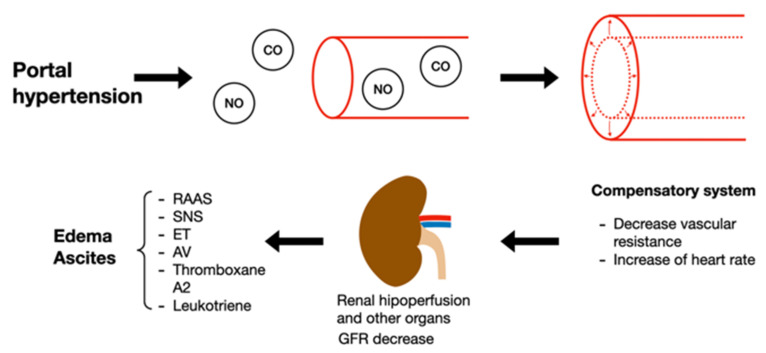
Pathophysiology of kidney damage induced by liver disorders. NO: nitric oxide; CO: carbon monoxide; GFR: glomerular filtration rate; RAAS: renin angiotensin aldosterone system; SNS: sympathetic nervous system; ET: endothelin; AT: arginine vasopressin.

**Figure 5 plants-11-00818-f005:**
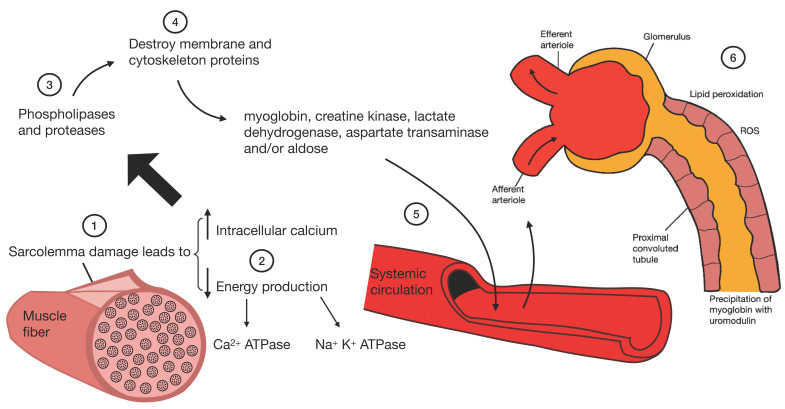
Pathophysiology of kidney damage induced by rhabdomyolysis.

**Figure 6 plants-11-00818-f006:**
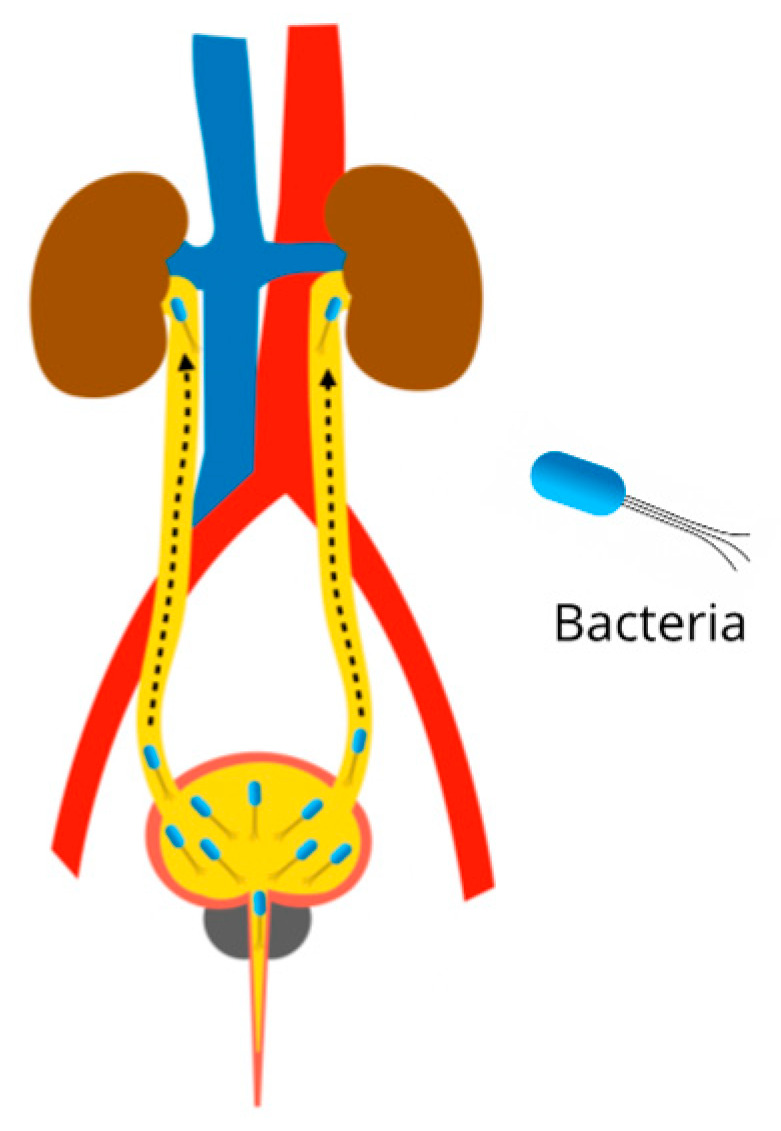
Pathophysiology of pyelonephritis due to urinary tract infections. Ascending infection from the bladder to the kidneys.

**Figure 7 plants-11-00818-f007:**
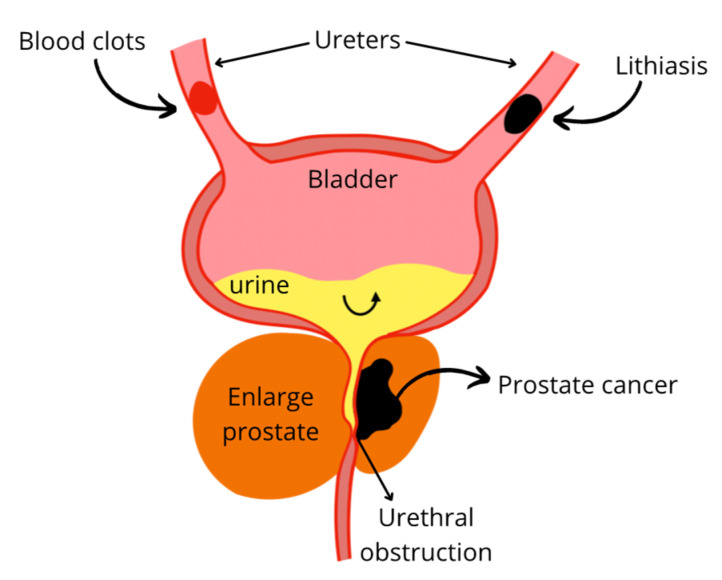
Urinary flow obstruction due to lithiasis, blood clots, or prostate enlargement.

**Table 1 plants-11-00818-t001:** Plants and their activity against pre-renal, intrinsic, and post-renal diseases.

Plant	Origin	Extract/Compounds	Model of Study	Disease	Mechanism of Action	Renal Effect	Reference
*Tinospora crispa*	Southeast Asia and Africa	Genkwanin	Albino Wistar rats + CCl_4_	Hepatic injury	Increase SOD activity	Pre-renal	[50]
*Cirsium vulgare* and *Cirsium ehrenbergii*	Amazon basin	Lupeol acetate	Male Wistar rats + CCl_4_	Hepatic injury	Prevent depletion of glycogen, antioxidant and anti-inflammatory effects	Pre-renal	[51]
*Suaeda vermiculata*	South Africa	Quercetin, quercetin-3-*O*-rutinoside, and kaempferol-*O*-(acetyl)-hexoside-pentoside	Male Sprague Dawley rats + CCl_4_	Hepatic injury	Decrease AST and ALT	Pre-renal	[52]
*Descurainia Sophia*	Europe and northern Africa	Dried seed ethyl alcohol Extract	Swiss albino mice + acetaminophen	Hepatic injury by NSAID	Reduce inflammation, swelling and necrosis	Pre-renal	[53]
*Eurycoma longifolia*	Indonesia	Standardized aqueous extract of the roots (“Physta” from the brand Biotropics)	Wistar rats + paracetamol	Hepatic injury by NSAID	Increase antioxidant enzymes, improves biomarkers of kidney function, and histopathology changes	Pre-renal	[54]
*Arbutus pavarii*	Libya	Gallic acid, epicatechin, dimeric forms of B-type Proanthocyanidins, quercetin, flavonoids, and phenolic acid	Disc Diffusion Assay	Hepatic injury by antibiotics	Exert bacteriostatic and bactericidal effect, against different methicillin-resistant *Staphylococcus aureus* strains	Pre-renal	[55]
*Punica granatum*	India	Dulcitol, loganin, bergenin, quercitrin, cosmosin, folic acid, khayanthone	Wistar rats + gentamicin	Antibiotics/infections	Improve kidney function biomarkers, exerted antioxidant activity, and ameliorated histological changes	Pre-renal and intrinsic	[56]
*Punica granatum*	India	Fruit peel ethanolic extract. Polyphenols.	Wistar rats + gentamicin	Antibiotic-induced liver and kidney damage	Protect the tissues against ROS-mediated oxidative damage and modulate the inflammatory response	Pre-renal and intrinsic	[57]
Cocoa (*Theobroma cacao*)	Mexico	Hydroalcoholic extract of Natural Forastero cocoa powder.	Zucker diabetic fatty rats	Type 2 DM	Decrease glucose levels	Pre-renal	[58]
*Coffea arabica*	Africa	Pulp aqueous extract	Wistar rats + high-fat diet	Type 2 DM	Raise catalase levels	Pre-renal and intrinsic	[59]
*Eysenhardtia polystachya*	Mexico	Methanolic bark extract	Mice + streptozotocin	Type 2 DM	Decrease oxidative stress	Pre-renal and intrinsic	[60]
*Anchomanes difformis*	Africa	Aqueous leaf extract	Wistar rats + fructose and streptozotocin	DM related with pancreas	Induce dissociation of Nrf2/keap, activating Nrf2, reduced oxidative stress	Pre-renal	[45]
*Hibiscus sabdariffa*	Asia, Africa, Central America	Calyx aqueous extract	Wistar rats + 30% sucrose	Metabolic syndrome	Increase the antioxidant systems including non-enzymatic and enzymatic effect	Pre-renal	[61]
*Hibiscus sabdariffa*	Asia, Africa, Central America	Calyx aqueous extract and dried powdered calyx Hibiscus acid, anthocyanins, chlorogenic acid	Humans with uncontrolled hypertension	Hypertension	Regulate blood pressure	Pre-renal	[46]
*Cichorium intybus*	Saudi Arabia	1,4- naphthalenedione, oleic acid, β-asarone, naphtho furanone, p-methoxycinnamate, hexadecanoic acid	Wistar albino rats + ISO-induced myocardial ischemia model	Hypertension	Improve the systolic function and increase the levels of LVEF and LVFS	Pre-renal and post-renal	[62]
*Gardenia jasminoides*	Asia	Geniposide	Wistar SHR, and Wistar Kyoto rats	Myocardial ischemia	Exert antioxidant activity and decreased CK-MB, AST, ALT, and MDA levels.	Pre-renal and post-renal	[63]
*Curcuma longa*	India	Curcumin	C57BL/6J mice Glycerol	Rhabdomyolysis	Reduce ROS, inflammation, and histopathology changes	Pre-renal	[64]
*Curcuma longa*	India	Curcumin	Wistar rats + doxorubicin	Nephrotoxicity	Increase enzymatic antioxidant activity	Pre-renal and intrinsic	[65]
Passion fruit (*Passiflora* spp.)	North America	Methanolic peel extract Gallic acid, Ellagic acid, Kaempferol and Quercetin glycosides	Albino rats + paracetamol	Nephrotoxicity	Keep urea and creatinine at normal levels	Pre-renal and intrinsic	[66]
*Euphorbia paralias*	Europe, western Asia, and northern Africa	Methanolic extract of aerial parts	Sprague-Dawley rats + thioacetamide	Nephrotoxicity	Reduce levels of urea and creatinine	Pre-renal and intrinsic	[67]
*Pistacia atlantica*	North Africa, Middle East, Iran, and Afghanistan	Leaf hydroethanolic extract	Wistar rats + gentamicin	Nephrotoxicity	Decrease levels of urea, creatinine, and uric acid	Pre-renal and intrinsic	[68]
*Costus afer*	Africa	Aqueous leaf extract	Wistar rats + cyclosporine	Nephrotoxicity	Decrease serum potassium and BUN levels	Pre-renal and Intrinsic	[69]
Cranberry (*Vaccinium* sp.)	North America	“Exocyan” brand natural cranberry extract. “Nutrican” brand cranberry dry extract.	Uropathogenic *Escherichia coli*	Urinary tract infections	Decrease *E. coli* adhesion, and reduce bacterial motility and biofilm formation	Post-renal	[70,71]
*Descurainia sophia*	Europe and northern Africa	Aqueous seed extract	Male Wistar rats + Ammonium chloride + ethylene glycol	Lithiasis	Decrease the deposition of calcium oxalate and amount of tissue damage	Post-renal	[72]
*Equisetum arvense*	Spain	Dry standardized extract of aerial parts. Alcoholic extract of sterile stems	Clinical trial Healthy male volunteers	Urinary retention andinfections	Diuretic action and effective against *Candida tropicalis, Candida glabrata, Candida albicans*, *Staphylococcus epidermidis*, *Streptococcus mutans* and *Staphylococcus aureus*	Post-renal	[73,74]
*Cynanchum wilfordii*	Korea	4-hydroxyacetophenone and 2,4-hydroxyacetophenone	Male Sprague-Dawley rats + testosterone	Benign prostatic hyperplasia	Decreased testosterone and DHT, via downregulation of androgen receptor 5α gene expression	Post-renal	[75]
Pumpkin (*Cucurbita pepo*)	Mexico	Oil-free hydroethanolic pumpkin seed extract.Phytosterols and fatty acids	Open mono-center trial men with symptomatic benign prostatic hyperplasia.	Benign prostatic hyperplasia	Decrease residual urine volume and nocturia.Inhibit 5α reductase and decrease DHT level.	Post-renal	[76,77]
Broccoli (*Brassica oleracea* var. Italica)	Italy	Broccoli Sprouts Powder from Natural Sprouts Company, LLC	TRAMP	Prostate cancer	Decrease HDAC expression, and decline the acetylation of histone H3 lysine 18 and H3K9	Post-renal	[78]

CCl_4_: tetrachloride; SOD: superoxide dismutase; AST: aspartate aminotransferase; ALT: alanine aminotransferase; NSAID: Non-steroidal anti-inflammatory; DHT: dihydrotestosterone; DM: diabetic Mellitus; DM2: diabetes mellitus type 2; ISO: isoprenaline; LVEF: left ventricular ejection fraction; LVFS: left ventricular fraction shortening; SHR: spontaneously hypertensive rats; CK-MB: creatinine kinase myocardial band; MDA: malondialdehyde; ROS: reactive oxygen species; TRAMP: transgenic adenocarcinoma of the mouse prostate; HDAC: histone deacetylated.

## Data Availability

Not applicable.

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
