# Peer review of "Nephroprotective Plants: A Review on the Use in Pre-Renal and Post-Renal Diseases"

_plants, 2022, doi:10.3390/plants11060818_

Round 1
Reviewer 1 Report
The review entitled "Nephroprotective plants: A review on the use in pre-renal and post-renal diseases" is a good compilation of medicinal plants used to treat pre and post roots of kidney diseases. I am seeing that there is some overlapping with recent published papers (for example: PMID:34655670; DOI:10.7324/japs.2021.1101016), which by the way were not mentioned in here. The authors must to state which is different and what is new in their review in comparison with other one. As well as cite them.
Also I find useful the description of the molecules or at least the chemical structures of molecules described in Table 1 and their effects if any as Nephroprotective agent.
I also feel that the conclusion are too vague and must be in future directions which plants are worth to work and why.
Other than that the review is good.
Author Response
Dear Reviewer
Thank you for your valuable comments. Please find below the detailed responses to your comments. The manuscript has been improved for reading according to additional suggestions, comments, and recommendations of reviewers, so we hope it meets the standard required by Journal Plants. The changes made have been highlighted with GREEN background in the manuscript.
Reviewer #1:
The review entitled "Nephroprotective plants: A review on the use in pre-renal and post-renal diseases" is a good compilation of medicinal plants used to treat pre and post roots of kidney diseases. I am seeing that there is some overlapping with recent published papers (for example: PMID:34655670; DOI:10.7324/japs.2021.1101016), which by the way were not mentioned in here. The authors must to state which is different and what is new in their review in comparison with other one. As well as cite them.
Author’s Response:
Thank you for your valuable comments and suggestions. We added the references in numbers 2 (DOI:10.7324/japs.2021.1101016) and 20 (PMID:34655670), and highlighted the new insight of the current review line 101-106.
Previous works already reviewed the usages of plants and phytochemicals as nephroprotective agents providing an important understanding of how extracts or single compounds interfere with molecular pathways to mitigate kidney diseases (2,20). However, they usually focused on intrinsic damage such as nephrotoxicity, omitting the pre-renal and post-renal factors, and to our knowledge, no classification of nephroprotective plants according to pre-, intra-, and post-renal diseases have been reported.
Also I find useful the description of the molecules or at least the chemical structures of molecules described in Table 1 and their effects if any as Nephroprotective agent.
Author’s Response:
Thank you for the suggestion. The chemical denomination of the molecules identified as active compounds were already reported table 1 when provided in the referenced papers. The chemical classification of molecules has been added where missing in the paragraphs where they are mentioned for greater understanding. It should be emphasized that there are few works in which an exact characterization of the molecular structures of the compounds present in the plant extracts that presented beneficial effects to mitigate pre- and post-renal diseases was made, therefore it is delicate to add such structures in the literature review work. Therefore, this review focused on the mechanism of action of the plants extracts on pre-renal and post-renal metabolic disorders, rather than seeking attribution of the effect to a specific compound since only a few studies accurately characterized the active compounds in the tested extracts. Although it was interesting to discuss the role of the compounds in the observed effects. Therefore, the compounds were classified without specifying their exact structure due to the lack of information in this regard.
I also feel that the conclusion are too vague and must be in future directions which plants are worth to work and why.
Other than that the review is good.
Author’s Response:
Thank you for your comment. The conclusion was reorganized and some parts were reformulated (lines 685-686, 704-707 and 711-712).
Reviewer 2 Report
Although the study is valuable, it has some shortcomings. Various situations should be considered that will increase the research value. The introduction should be modified with a clear understanding for readers and should be rearranged to be more understandable.
Typos should be corrected. The article should be accepted after minor revision.
Author Response
Dear reviewer,
Thank you for your valuable comments. Please find below the detailed responses to your comments. The manuscript has been improved for reading according to additional suggestions, comments, and recommendations of reviewers, so we hope it meets the standard required by Journal Plants. The changes made have been highlighted with GREEN background in the manuscript.
Reviewer #2:
Although the study is valuable, it has some shortcomings. Various situations should be considered that will increase the research value. The introduction should be modified with a clear understanding for readers and should be rearranged to be more understandable.
Author’s Response:
Thank you for your valuable comment. The introduction was modified for better understanding.
Particularly:
- Chronic and acute kidney disease was defined lines 52-54
- Characteristics of kidney disease and clinical indications for their diagnosis were removed for the first paragraph, and the information was used to complement the second paragraph lines 63-66.
- The third paragraph was moved above figure 1.
- The fourth paragraph was separated line 99 and a highlight of the new insights of the manuscript compared to other reviews was added lines 100-106.
- Sentences were reformulated lines 74, 76-79, 106-108, and 121-122.
Typos should be corrected. The article should be accepted after minor revision.
Author’s Response:
Thank you for your valuable comment. Typos, punctuation, and grammar were revised throughout the manuscripts.
Similarly, the conclusion was re-edited for greater clarity for the reader.
Reviewer 3 Report
This review focuses on the diversity of plants that have scientifically demonstrated their biological activity against renal disorders in Pre-renal factors and Post-renal. The mechanisms of action of the plant products they regulate are also considered. Moreover, it provides a better understanding of how plants act as molecular modulators to alleviate various health disorders that may indirectly or subsequently lead to the development of kidney disease.
I have no comments on the structure and the information included and propose that the review be adopted in its current form.
Author Response
Dear reviewer,
Thank you for your review. Please find below the detailed responses to your comments. The manuscript has been improved according to additional recommendations, suggestions, and comments of reviewers, so we hope it meets the standard required by Journal Plants. The changes made have been highlighted with GREEN background in the manuscript.
Reviewer #3:
This review focuses on the diversity of plants that have scientifically demonstrated their biological activity against renal disorders in Pre-renal factors and Post-renal. The mechanisms of action of the plant products they regulate are also considered. Moreover, it provides a better understanding of how plants act as molecular modulators to alleviate various health disorders that may indirectly or subsequently lead to the development of kidney disease.
I have no comments on the structure and the information included and propose that the review be adopted in its current form.
Author’s Response:
Thank you for your review and consideration. Some changes highlighted in green were made to improve the manuscript
Reviewer 4 Report
Manuscript No. plants-1628266
„Nephroprotective plants: A review on the use in pre-renal and post-renal diseases” for Plants
Comments:
- Paragraph 4.1.2. Please specify that the range of concentrations considered as hypertension also depends on the age of the patient.
- Lines 202, 232, 378, 560. Please change the letter k to (kappa symbol) in the name of NF-kB
- Please standardize the titer records. Either with a space after a numeric value or without a space.
- Paragraph 4.2.1. Please give examples of the most common uropathogens at the beginning of this section.
- Paragraph 4.2.3. In my opinion, this chapter should be divided into two subsections. Please distinguish between benign prostatic hyperplasia and prostate cancer. Certainly, both types of hypertrophic changes will be treated differently.
Author Response
Dear reviewer,
Thank you for your review. Please find below the detailed responses to your comments. The manuscript has been improved according to additional recommendations, suggestions, and comments of reviewers, so we hope it meets the standard required by Journal Plants. The changes made have been highlighted with GREEN background in the manuscript.
Reviewer #4:
Comments:
- Paragraph 4.1.2. Please specify that the range of concentrations considered as hypertension also depends on the age of the patient.
Author’s Response:
Thank you for your valuable comment. Please see lines 270-272: Information that independently of the range considered to define hypertension the pharmacological treatment depends on age and comorbidities were complemented and two references were added.
- Lines 202, 232, 378, 560. Please change the letter k to (kappa symbol) in the name of NF-kB
Author’s Response:
Thank you for your valuable comment. NF-kB has been replaced by NF-κB throughout the manuscript. Please see lines 212, 241, 387, 568.
- Please standardize the titer records. Either with a space after a numeric value or without a space.
Author’s Response:
Thank you for your valuable comment. Titer records were homogenized throughout the manuscripts.
- Paragraph 4.2.1. Please give examples of the most common uropathogens at the beginning of this section.
Author’s Response:
Thank you for your valuable comment. Please see lines 580-582: the uropathogens which were mentioned in the referenced works were listed to complement the section.
- Paragraph 4.2.3. In my opinion, this chapter should be divided into two subsections. Please distinguish between benign prostatic hyperplasia and prostate cancer. Certainly, both types of hypertrophic changes will be treated differently.
Author’s Response:
Thank you for your valuable comment. The section was divided into two subsections as recommended. Benign prostatic hyperplasia was further defined and plant-based treatments were reviewed and referenced (lines 643-664). The cited papers were also added in table 1.
Round 2
Reviewer 1 Report
The review have been improved notably and now can be accepted